# Ketogenic Diet, Physical Activity, and Hypertension—A Narrative Review

**DOI:** 10.3390/nu13082567

**Published:** 2021-07-27

**Authors:** Domenico Di Raimondo, Silvio Buscemi, Gaia Musiari, Giuliana Rizzo, Edoardo Pirera, Davide Corleo, Antonio Pinto, Antonino Tuttolomondo

**Affiliations:** Department of Promoting Health, Maternal-Infant. Excellence and Internal and Specialized Medicine (Promise) G. D’Alessandro, University of Palermo, 90100 Palermo, Italy; silvio.buscemi@unipa.it (S.B.); gaiamusiari@gmail.com (G.M.); giulianarizzo@yahoo.it (G.R.); edoardo.pirera95@gmail.com (E.P.); davidecorleo@gmail.com (D.C.); antonio.pinto@unipa.it (A.P.); bruno.tuttolomondo@unipa.it (A.T.)

**Keywords:** ketogenic diet, blood pressure, essential hypertension, physical activity, exercise, aerobic capacity

## Abstract

Several studies link cardiovascular diseases (CVD) with unhealthy lifestyles (unhealthy dietary habits, alcohol consumption, smoking, and low levels of physical activity). Therefore, the strong need for CVD prevention may be pursued through an improved control of CVD risk factors (impaired lipid and glycemic profiles, high blood pressure, and obesity), which is achievable through an overall intervention aimed to favor a healthy lifestyle. Focusing on diet, different recommendations emphasize the need to increase or avoid consumption of entire classes of food, with only partly known and only partly foreseeable consequences on the overall level of health. In recent years, the ketogenic diet (KD) has been proposed to be an effective lifestyle intervention for metabolic syndrome, and although the beneficial effects on weight loss and glucose metabolism seems to be well established, the effects of a prolonged KD on the ability to perform different types of exercise and the influence of KD on blood pressure (BP) levels, both in normotensives and in hypertensives, are not so well understood. The objective of this review is to analyze, on the basis of current evidence, the relationship between KD, regular physical activity, and BP.

## 1. Methodology of Literature Search

### 1.1. Data Sources and Search

A comprehensive literature search was carried out in the MEDLINE database (search terms: “ketogenic diet” + “history”, “ketone bodies”, “physical activity”, “aerobic training”, “endurance”, “anaerobic training”, “hypertension”, “cardiovascular diseases”, “blood pressure”, “endothelial dysfunction”). The search has been restricted to papers published in English without time limit. The authors sought literature by examining reference lists in original articles and reviews. We have included in this review only systematic reviews, metanalyses, randomized trials, and randomized controlled trials, selecting studies in which the intervention was ketogenic diet or very low carbohydrate ketogenic diet and one of the main objectives was to examine the effects on exercise capacity and/or on blood pressure (BP) levels.

### 1.2. Data Analysis

Each author involved independently evaluated the results of the literature research, extracting the most pertinent knowledge whilst others verified the accuracy and completeness of the extracted data. Each author made a judgement as to whether the search results were different or confounding in order to release a complete overview of the field.

## 2. Introduction

A ketogenic diet (KD) is a high-fat (providing a range of 55 to 90% calories as fat), adequate protein (accounts for 30–35% of the daily caloric requirement supplied; minimum of 1 g/kg of protein), low-carbohydrate diet (only 5–10% of total calories are provided by carbohydrates, less than 50 g/day) [1]. The different availability of substrates supplied to the organism by the diet influences the metabolism and drives it to use different energy substrates according to both quantity and quality of nutrients consumed in the specific dietary regimen. This particular type of diet, designed to increase production of ketones by simulating the metabolic changes of starvation [2], has shown increasing interest from both the scientific community and patients since the early 1920’s, when the KD was successfully used as a therapy for intractable childhood epilepsy [3], has its cornerstone on the voluntary deficiency in carbohydrate intake leading the body to a rapid depletion of glycogen reserves; given the persistent unavailability of carbohydrates, the body turns to different metabolic pathways: gluconeogenesis and ketogenesis [1]. This "metabolic shift" is potentially very beneficial because ketone bodies produce more adenosine triphosphate in comparison to glucose and can be easily utilized for energy production by the heart, muscle tissue, brain, and kidneys (but not for red blood cells and the liver) [2]. This is basically the opposite effect to what happens in states of excess of carbohydrate consumption, when we may observe an elevation in glucose and insulin levels with a subsequent anabolic state in which fatty acids are driven towards storage rather than utilization.

In fact, it is probably more accurate to talk about “ketogenic diets”: there is not a registered unique specific protocol for the “KD”. Different diet methodologies are offered to patients depending on (i) level of carbohydrate restriction, (ii) protein contribution, (iii) quality of fat (animal and/or vegetable). It is therefore clear that the consequences on the metabolism as briefly outlined before can be variable in relation to a different approach more or less “fundamentalist” to KD. Moreover, these ketogenic diets should be considered part of the larger group of low carbohydrate diets (LCD), including in this term a very heterogeneous group of nutritional regimens, without a univocal definition [4], which have as a key common denominator a low content of carbohydrates. Some examples of LCDs are the Atkins diet [5], the Zone diet, the South Beach diet, and the Paleo diet. [6]. Given that many epidemiologic analyses conducted on different large groups of subjects have established that the average daily intake of macronutrients is at least 45% provided by carbohydrates [6], the definition of LCD should be attributed to a diet that provides between 50 and 150 g of carbohydrates per day (equivalent to a percentage > 10% and <30%) while we can talk of KD for a diet that provides a <50 g per day of carbohydrates (equivalent to a percentage < 10%) although very often only a daily intake < 20 g is allowed. The lower the quantity of carbohydrates supplied in the diet, the higher will be the formation of ketones and therefore the “ketogenicity” of the diet [1,2,6].

In view of the intrinsic heterogeneity of the topic addressed in this review, we are going to refer primarily to KD, extending our analysis in relation to the evidence available also to all those studies that, even without clearly defining the proposed diet as ketogenic, have tested a diet in which a quantity of carbohydrates < 50 per day was provided, since it is often impossible to make a clear distinction between these different dietary approaches.

In consideration of its encouraging effects on carbohydrate metabolism and glucose levels, the scientific community’s interest in KD was headed towards finding methods to combat the worsening obesity epidemic [7]. Obesity, as well as many other diseases like diabetes mellitus and cardiovascular diseases (CVDs), is a condition with several contributing causes including poor dietary habits and sedentary physical activity behaviors [8,9]. Of note, data have been reported suggesting that some LCDs may also have unfavorable effects on cardiovascular (CV) and endothelial function [5,10]; this confirms the need to study all the short-term and long-term effects exerted by LCDs and KDs more in-depth in order to determine whether these diets may be safely implemented in patients at high CV risk or in subjects having already reported a previous vascular event [5,10].

KD substantially induces a metabolic framework that mimics starvation: during a short-limited period of nutrient deprivation or low carbohydrate availability, the primary source of carbohydrate reserve is glycogen, a branched polymer of glucose serving as a store of energy in times of nutritional sufficiency for utilization in times of need [11], which provides only 12- to 14-h energy reserve [12]. Therefore, when fasting is prolonged and glycogen reserves are depleted, in order to supply the unavailable dietary glucose, the gluconeogenesis process is stimulated, and the primary carbon skeletons required for the synthesis of glucose come from lactic acid, glycerol, and the amino acids alanine and glutamine [2]. When the endogenous production of glucose by gluconeogenesis remains too low to cover the body’s glucose needs, ketone bodies will be produced as an alternative to glucose. Then, the main source of energy becomes dietary fat and then fat stored in adipose tissue which is metabolized in hepatocyte mitochondria in ketone bodies. Fatty acids are transported into mitochondria, then undergo the β-oxidation process, which results in the production of acetyl-CoA. Under conditions of reduced glucose availability (prolonged fasting, KDs), acetyl-CoA undergoes a series of biochemical modifications that result in the formation of acetone, acetoacetate, and β-hydroxybutyric acid [1,13,14,15,16].

A KD is usually followed for a minimum of 2 to 3 weeks up to 6 to 12 months. continuing KD for an excessively prolonged period (beyond six months) is generally not recommended unless under very close supervision and periodic clinical re-evaluation [10].

KD (in some experimental work you can find the expression “very low carbohydrate diet” (VLCD) or “very low carbohydrate ketogenic diet” (VLCKD) these terms being used as an equivalent of KD [17]) has been shown to be effective in the short to medium term (three to six months) in helping control lipid profile and as a tool to counteract obesity, leading to a significant decrease in weight, body mass index (BMI), and fat mass, although to date scarce data are available regarding the patient’s ability to maintain weight loss over time [17,18,19]. Moreno et al. [18], using a very low carbohydrate ketogenic diet (<50 per day of carbohydrates), reported a selective reduction in visceral fat measured by a specific software of dual-energy x-ray absorptiometry (DEXA)-scan (−600 g vs. −202 g using a standard low-calorie diet; *p* < 0.001) [18].

Nevertheless, of this wide range of beneficial effects, various reports suggest short-term and long-term potential adverse effects related to the adoption of KD. One of the main short-term side effects of the KDs is the so-called “keto flu” [20], also often referred to as “keto-induction” or “keto-adaptation” [21,22], a cluster of transient symptoms generally reported as occurring within the first few weeks of KD, predominantly constipation, headache, halitosis, muscle cramps, diarrhoea, vomiting, and general weakness [20]. To date the cause of the occurrence of keto flu is not fully explained and very few authors have addressed this condition [20,22]. The risk of occurrence of keto flu is reported to be higher when the caloric intake is too low or the diet includes periods of total fasting that are particularly prolonged and recurrent [21] and the main hypothesis regarding the cause is the increased urinary sodium, potassium, and water loss in response to lowered insulin level as well as the altered glucose bio-availability for the brain [20].

Another unfavorable metabolic disarrangement linked to prolonged KD is a substantial rise in low density lipoprotein (LDL) cholesterol levels. This finding, along with the report of a KD-induced endothelial disfunction diet [5] and other questions regarding the overall CV health during and after prolonged phases of KD have led many experts to clearly express concerns regarding its long-term effects, especially towards CV function [23].

In this regard, it must be emphasized that the overall effect of a KD on cardiovascular wellbeing, but also the overall health effects, depends not only on the amount and type of carbohydrate intake but also on the origin of the proteins provided (plant proteins rather than red meat or fish) and the type of fat intake (butter or other animal fats rather than olive oil and nuts). A good experimental demonstration of this important rationale was provided in 2010 by Fung et al. reporting that the consumption of a vegetable-based, as opposed to an animal-based, low-carbohydrate diet can be associated with a lower risk of all-cause and CVD mortality, suggesting that the health effects of a low-carbohydrate diet may depend on the type of protein and fat provided rather than by the altered proportion of nutrients supplied itself. [24]. It is therefore likely that a possible unfavorable effect of KD on the LDL level is not attributable to the diet per se, but rather to the type of lipids that the diet encourages one to consume.

In addition to its role in determining modifications in metabolic substrates, KD has a role in modulating mitochondrial renewal (via mTOR pathways), neurotransmission, oxidative stress, and inflammatory mechanisms. [16] The final effect is a better neuronal resistance and adaptive ability to metabolic stress and challenges. [16,25,26].

In view of these premises and of the growing interest that KDs are gaining in an increasingly large audience of potential patients, this review aims to investigate two aspects that we believe are extremely relevant for individuals approaching this type of diet, namely (1) how much the ability to exercise is influenced by the different bioavailability of metabolic substrates seen during the KD, and in particular by the scarcity of glucose, which is the fuel used by the muscle to support many of the physical efforts, especially those of higher intensity and short duration [27], and (2) what kind of benefit we can expect from this type of diet on blood pressure, since pathophysiologically obesity, altered glucose metabolism, and altered blood pressure control are closely interconnected.

## 3. Ketogenic Diet and Physical Activity

Physical Activity (PA) can be defined as any bodily movement produced by the contraction of skeletal muscles that results in a substantial increase in caloric requirements over resting energy expenditure [28]. During PA, muscles rely on their active contraction on three major pathways, i.e., the phosphagen system (anaerobic alactacid), the lactic acid system (anaerobic lactacid), and the aerobic system. These three pathways, whose goal is that of ensuring ATP availability throughout the contraction time, are preferentially enabled in relation to the duration and the intensity of exercise [29]. More specifically, the phosphagen system and the lactic acid system can be referred to as the “anaerobic system”. The key mechanisms to first answering muscles’ energy requirements are (i) the collection of stored and already disposable ATP in the cell, (ii) the activation of the phosphagen system that consists of the splitting of the high-energy phosphagen and phosphocreatine (PCr) [30]. if these mechanisms are not able to provide adequate metabolic support to the contracting muscle, a further metabolic pathway takes over: the non-aerobic breakdown of carbohydrate, obtained from hepatic and muscle glycogen storage, degraded into pyruvic acid and then lactic acid through glycolysis [31]. The third, aerobic or oxidative metabolism, involves the combustion of carbohydrates and fats, and only in a few cases of proteins, in the presence of oxygen [32].

The pattern of activation of these three different pathways depends on the type of exercise chosen: in high intensity, short-term exercise, muscle contraction will rely upon anaerobic pathways (the phosphagen system and the lactic acid system), whilst in low-to-moderate intensity endurance exercise their contraction will only initially rely upon the latter and then switch to aerobic metabolic pathways, fueled by liver and adipose tissue which provide a more stable, less finite source of energy (e.g., adipose tissue). Since the pattern of activation of these integrated processes is variable as well as the main source of energy used, it is reasonable to think that athletes could benefit from a different type of dietary regimen depending on their main PA program.

Endurance training (ET) is a type of exercise usually performed at constant intensity, with the main purpose of progressively increase the “anaerobic threshold”, i.e., the limit above which the organism begins to use the anaerobic metabolism to restore the depleted ATP at the cost of accumulating lactate production [33]. Particularly for submaximal or maximal intensity exercises, the extremely rapid increase in the muscle’s demand for oxygen cannot be fulfilled immediately by the aerobic system thus creating a temporary “oxygen deficit” during which, as previously stated, the phosphagen system and the lactic acid system are the major suppliers of ATP synthesis. [34] Once the deficit is filled, a series of coordinated metabolic processes take place to preserve the supply of exogenous substrates. The liver has the primary role of sustaining blood glucose levels both via glycogenolysis and gluconeogenesis and can produce ketone bodies from elevated serum concentrations of fatty acids that come from the lipolysis of adipose tissue (activated by beta-adrenergic stimulation during exercise) [35]. In this scenario, where fat from adipose tissue is considered a steady supply of energy and ketone bodies are considered an alternative or supplemental fuel source to sustain endurance exercise, evidence suggests that KD may be advantageous for ET by promoting fat use, rather than carbohydrates, for fuel [29]. KD may provide a fine gain by stimulating both fat usage and ketone body production, thus sustaining low-to-moderate, long-duration exercise and oxidative metabolic processes [29].

Notably, ketone bodies, whose production is triggered by the adoption of ketogenic diet, seems to have an increased efficiency in generating metabolic energy compared with glucose and fatty acids [1]. This better efficiency in energy production could be determined by ketone bodies’ ability to generate more power while consuming the same amount of oxygen (i.e., requiring less oxygen per mole of carbon during their oxidation) [36] and this could facilitate a higher power output at the same VO2 (maximal oxygen consumption), thus increasing maximal performance [35].

These assumptions have found only partial confirmation in experimental results; Volek et al. [37], switching ultra-endurance athletes from a high-carbohydrate (684 g/day) to a low-carbohydrate diet (82 g/day) for an average 20 months, found that the chronic keto-adaptation was associated with a greatly increased capacity to oxidate fat during exercise while maintaining normal skeletal muscle glycogen concentrations. In this study, during submaximal exercise, fat contributed 88% of total energy expenditure in the low carbohydrate group vs. 56% in the high carbohydrate group and keto-adapted subjects had ~3-fold higher levels of circulating beta-hydroxybutyrate and total ketones at rest, during exercise, and in recovery [37]. This low carbohydrate diet is not a ketogenic diet in the strict sense, but given the metabolic adaptations evidenced during endurance training, it is likely that they can be extended to less trained subjects and to even more ketogenic diets, although suitable experimental confirmation is needed to substantiate the magnitude of the metabolic adaptation in other groups of subjects.

Volek et al. [37] did not compare exercise performance between the two groups whilst Burke et al. [38], in a similar experimental context, reported that keto-adaptation to a LCKD (<50 g/day of carbohydrates) negated performance benefits in elite endurance athletes, in part due to reduced exercise economy. Most of the studies that examined the effect of LCDs or KD on ET performance in humans reported similar results: effects on body composition (decrease in body weight and fat mass with slight variations in lean mass) and in ketone utilization for fuel, but little or no improvement in exercise performance [38,39,40,41,42,43,44,45,46,47,48].

Although the available evidence to date is extremely mixed, partially due to the heterogeneity across studies and/or variability in athletes’ individual characteristics, and because of the small sample size of the population studied, VLCKDs seem to be ineffective at producing significant variations in exercise performance, or rather, in some datasets, lead to an impaired exercise efficiency, particularly at >70% VO^2^max, as evidenced by increased energy expenditure and oxygen uptake [38,45], whereas during long endurance exercise at moderate intensity (between 50% and 70% VO^2^max), lower plasma lactate concentration and increased aerobic capacity has been reported [46].

In conclusion, the hypothesis that consuming a KD may enhance performance during ET by promoting a shift in more efficient metabolism balance is not confirmed by available research. Several mechanisms have been implicated to explain the potential for mixed and/or detrimental effects, including changes in fuel economy, production of certain metabolic byproducts, and reduced energy intake [48]. Fat oxidation requires greater oxygen consumption due to the increased oxygen demands during fatty acid metabolism versus carbohydrate metabolism; some KD-related catabolites such as ammonia may promote fatigue by influencing the central nervous system, a significant reduction in caloric intake and body weight (often due to the KD-induced premature satiety) in several subjects may negatively impact the mental approach to physical performance as well as muscular and bone health, recovery time, and general exercise performance [48].

Figure 1 outlines the main effects of KD on metabolism during exercise.

Resistance Training (RT), often associated with the use of weights, in contrast to ET, has as its main objective the improvement of strength. RT is primarily anaerobic, even while training at a lower intensity (training loads of 0–40% of one-repetition maximum [one rep maximum or 1RM], i.e., the maximum amount of weight that a person can possibly lift for one repetition) [49].

For anaerobic performance, the evidence suggests that VLCKDs have a detrimental effect when compared to a high-carbohydrate diet with a reported significant lower peak and mean power exerted [50,51]. Furthermore, some studies have shown a significant decrease in skeletal muscle thickness [42] and in lean body mass [51], this could be determined by the fact that short-term, high intensity exercise relies on ATP resynthesis via anerobic pathways in which glycogen utilization is the primary source of energy (see Figure 1), KD is, in fact, known to promote fatty acid utilization as an energy source rather than glucose [6] (see Figure 1).
Figure 1Effects of Ketogenic Diet on the availability of metabolic substrates in relation to the type of exercise [38,39,40,41,42,43,44,45,46,47,48,50,51].
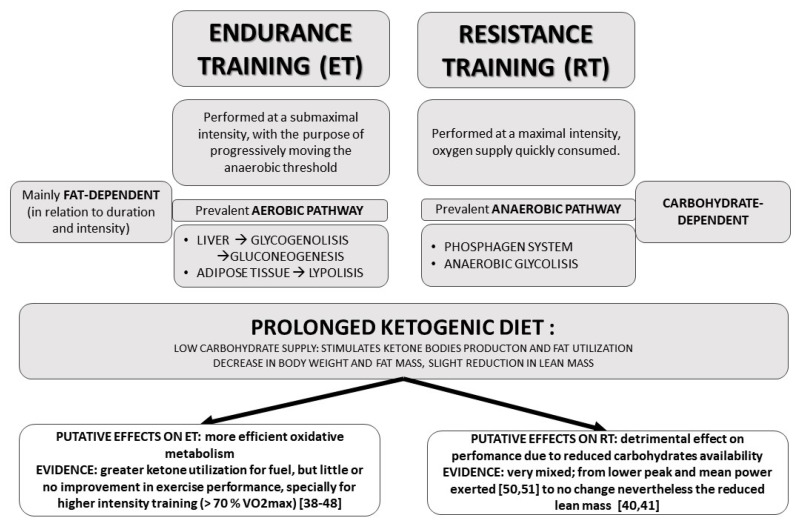



During endurance training, the growing ATP request is sustained by substrates deriving from organs able to provide a more stable source of energy (i.e., liver and adipose tissue). The initial steps of this type of exercise still cause a transitory oxygen deficit that causes the organism to initially leverage anaerobic pathways to answer the need for ATP resynthesis. In this scenario, a ketogenic diet could putatively sustain aerobic pathways by stimulating fat usage and ketone body production, nevertheless no data to date shows significant improvements in exercise performance in subjects following the KD. Resistance Training is based primarily on short-term, high intensity bouts of exercises often involving weights. This type of exercise is supplied by glycolitic anaerobic pathways, relying for energy production on “finite” sources of energy. Since a ketogenic diet disfavors these pathways by depleting its substrates’ disposability, findings suggests that its role could be detrimental rather than beneficial. See text for more info.

## 4. Effects of Ketogenic Diet on Blood Pressure Levels

Since essential hypertension is estimated to affect about 40% of the population globally [52], it is an independent risk factor for heart disease, stroke, and chronic kidney disease [53] and is a main feature of the metabolic syndrome (MS), along with obesity, diabetes, and dyslipidemia, understanding the effects of a specific diet on blood pressure values is extremely important in clinical terms [54].

Starting from the first pioneering results obtained with the Dietary Approaches to Stop Hypertension (DASH) diet [55], a dietary plan in which some simple dietary guidelines (eat more fruits, vegetables, and low-fat dairy foods, eat less foods with low saturated fat, eat more whole-grain foods, fish, poultry, and nuts, limit sodium, sweets, sugary drinks, and red meats) have been able to help patients to lower their blood pressure values, demonstrating that specific dietary regimens can significantly reduce blood pressure levels, many studies have shown that while a high intake of carbohydrates seems ineffective (or unfavorable) against the features of MS [56,57,58], dietary regimens restricting carbohydrates seem to be more effective at controlling all MS features [59,60,61].

However, although KD has been proposed to be an effective lifestyle intervention for MS, its effects on hypertension and in general on blood pressure values have yet to be deeply investigated [53]. To date, there is no randomized controlled clinical trial designed to evaluate KD in comparison with other specific diets that are based on different assumptions, whose main objective is to evaluate the effects of the complex metabolic alterations induced by KD on blood pressure levels in a large population. The few studies available have often been carried out on very select categories of subjects, such as the morbidly obese. The types of diet suggested are extremely diverse, as already pointed out, and the study designs have often compared very different plans. It is certain that the control of body weight and the reduction of fat mass per se determines an improvement in the blood pressure profile in each subject regardless of how this objective is achieved [52]. For all these reasons and because of the limited sample size that is common to almost all available studies, the data are not univocal (see Table 1).

Different meta-analyses during recent decades have tried to provide some indications to patients. In 2003, Bravata et al. [73], comparing available data on low carbohydrate, ketogenic, or higher protein vs. higher fat diets, found overall no change in systolic blood pressure values after diet in participants receiving either lower or higher carbohydrate diets; due to the low-quality data of the available studies, the authors’ statement was that there is insufficient evidence to make recommendations for or against the use of low-carbohydrate diets, particularly among participants older than 50 years, for use longer than 90 days, or for diets of 20 g/d or less of carbohydrates [73]. In 2009, Hession et al. [74], comparing the data of thirteen randomized controlled trials of low-carbohydrate diets vs. low-fat/low-calorie diets, found significant differences between the groups for weight, high-density lipoprotein cholesterol, triacylglycerols, and systolic blood pressure (SBP) favoring the low-carbohydrate diet (mean difference of SBP: 2.19 mmHg (*p* = 0.05), mean difference of DBP: 0.76 mmHg (*p* = 0.37)). There was a higher attrition rate in the low-fat compared with the low-carbohydrate groups, suggesting a patient preference for a low-carbohydrate/high-protein approach [74]. In 2012, a meta-analysis performed on 17 controlled trials involving 1141 obese patients [75], showed an overall effect of LCDs on BP estimated at −4.81 mm Hg (95% CI −5.33/−4.29) for systolic BP and −3.10 mm Hg (95% CI −3.45/−2.74) for diastolic BP. The authors also reported a global benefit on cardiovascular health, including amelioration of lipid-related indicators with an unexpected increase in high-density lipoprotein (HDL) cholesterol [75].

Undoubtedly, the improvement in blood pressure values obtained through the diet is largely mediated by the reduction in body weight, fat mass, and by the indirect effects resulting from the improved control of cardiovascular risk factors [52,76]; it can therefore be deduced that the greater the ability of a diet to reduce body weight, reduce fat mass, and improve the cardiovascular risk profile of an individual, the greater the potential of the diet to ensure more effective control of blood pressure values. Many are the elements that interlink cardiovascular wellness, blood pressure levels, and diet: one of the most important linkages among all of these is endothelial dysfunction [77]. Obese or overweight subjects are likely to have an obesity-induced, low-grade, pro-inflammatory state able to create per se an endothelial dysfunction state and thus a favorable substrate for the development of both cardiovascular and dysmetabolic diseases [78]. Therefore, in order to analyze the relationship between KDs and hypertension, it is important to briefly discuss the effects of KDs on other CV risk factors. KD putatively exerts its effects on the CV system, both indirectly through weight loss as well as directly through more specific mechanisms. As for the first, it is well known that an excess fat mass worsens most cardiovascular disease (CVDs) risk factors, such as dyslipidemia, hypertension, insulin resistance, and systemic inflammation [79]. Thus, KD, especially those very low in carbohydrates, by inducing rapid weight loss grant various beneficial effects on the pivotal risk factors for CVDs. [80].

The processes through which KD could directly influence the CV system are still to be understood. Many of the beneficial effects of calory restriction appear to be due to ketosis. Myocardium is the highest ketone body consumer per unit mass [81] and ketone bodies seem to be involved in the modification of specific nutrient-responsive pathways, including a series of epigenetic modification, i.e., covalent modifications (lysine acetylation, methylation, and hydroxybutyrylation), suggesting that ketones are dynamic regulators of chromatin architecture and gene transcription [82] For instance, β-hydroxybutyrate has been shown to suppress sympathetic nervous system activity and to reduce heart rate and total energy expenditure by inhibiting short chain fatty acid signaling through G protein-coupled receptor 41 (GPR41) [83]. Pre-clinical studies have also demonstrated that β-hydroxybutyrate blocks NLRP3 inflammasome [84], thus supporting the direct anti-inflammatory role of very low carbohydrate KD beyond its beneficial effects on metabolic parameters [85].

Evidence has shown that cardiac ketone oxidation is increased in the failing heart and it is likely that increased ketone oxidation can maintain cardiac energy supply and CV fitness in a situation with limited energy production such as diabetes mellitus or heart failure; ketone bodies may therefore be an adaptive and compensatory response to the impaired glucose metabolism in the diabetic heart. [86]. These findings suggest intriguing new perspectives, such as a putative beneficial use of ketone bodies to provide metabolic support to the failing heart, although further studies need to be performed, as very recently highlighted [87].

## 5. Conclusions

The growing interest in KDs, and in LCDs more generally, has led to several clinical studies that have tried to highlight various new pathophysiological aspects of this dietetic approach. There is not a univocal definition of LCD in the literature, hence it is very difficult to compare the results of different studies because many of them use different types of low-carbohydrate diets often not providing information on the specific content. Many of the studies have been conducted over a strict cohort of the population: obese or diabetic patients and, as for the studies focused on exercise performance, trained athletes. This gives a narrow interpretation of the results.

Nevertheless, of these limitations, KDs have been demonstrated to grant a clear beneficial impact on body composition and glucose/insulin homeostasis for obese and/or type 2 diabetes mellitus patients. However, some aspects of KDs and LCDs are still to be understood as they could be associated with long-term compliance to these dietary regimens, which are not so comfortable to follow. KDs are not always considered palatable by patients; it is often necessary to be very careful about micronutrients, providing adequate supplements to avoid nutritional deficiencies; KDs require strict compliance because, in order to maintain metabolic changes, a state of ketosis must always be preserved; the limitations and the sacrifices required can have an impact on the patient’s social life.

Due to the evidence shown, it is clear that many aspects related to the long-term consequences of KD need to be clarified, and the urgency appears to be greater the greater the number of people who approach these types of diets with confidence, often not being properly informed of the potential risks associated with LCDs, probably larger than those that can be encountered with diets based on different schemes.

Given the intrinsic characteristics of the diet and the common concomitance of different lifestyle interventions in the same subject, the relationship between KD and exercise performance need to be well ascertained. The availability of metabolic substrates for muscle contraction is strongly influenced by KDs. All individuals, both healthy and those carrying one or more chronic diseases, who choose to follow this type of diet should be informed about the potential consequences that KD may have on exercise capacity. As we have shown, different types of exercise, different duration and intensity, as well as the biometric characteristics of the patient condition the impact of KDs on exercise capacity. RT, which is more profoundly dependent on immediate glucose availability, seems to be negatively affected by KD as well as by all low carbohydrate diets. High intensity ET, particularly at >70% VO^2^max, seems also to be impaired due to a lower exercise efficiency with increased energy expenditure.

In this review, we also discuss the influence of KD on blood pressure levels. Since this dietary regimen is often recommended in obese diabetic subjects, who very frequently have abnormal blood pressure levels, researchers should, in our opinion, focus particularly on the definition of the short- and long-term effects of KDs on BP values in both normal and hypertensive subjects.

Available data seem to suggest that KDs are able to provide a reduction in blood pressure values but do not induce significantly different changes compared to non-ketogenic diets. These data suggest that the antihypertensive effects of KDs are not directly related to the precise metabolic consequences induced by ketosis but rather indirectly due mainly to weight loss.

The data on which we base these conclusions suffer from the limitations we have already pointed out. There is a great need for randomized controlled clinical trials designed ad hoc to provide certain findings that can be considered valid for all.

## Figures and Tables

**Table 1 nutrients-13-02567-t001:** Summary of studies addressing the effects of diets that supplied a daily intake of carbohydrates < 50 g on blood pressure levels.

Ref	Year	Methodology	Population	Patients	Intervention	Duration	Main Results	Other Results	Adherence
Samaha et al. [62]	2003	Randomized controlled trial	Age > 18 years and body mass index at least 35	132	Low-CHO (<30 g/day) vs. low-fat diet	6 months	Not significant overall or between group changes in BP (mean SBP −1.0 mmHg, mean DBP −3.0 mmHg vs. low-fat diet).	No patient needed change of antihypertensive therapy during the study	Less drop-out in low-CHO group; *p* = 0.03 at third month
Foster et al. [63]	2003	Randomized controlled trial	Obese men and women	63	Low-CHO (20 g/day), high-protein diet vs. low-calories, high-CHO, low-fat diet (“conventional”)	12 months	Not significant overall or between group changes in BP (mean SBP −2.7 mmHg, mean DBP −0.1 mmHg vs. “coventional” diet).		Level of adherence reported as “poor” in both groups
Dansinger et al. [64]	2005	Randomized Trial	Overweight or obese adults aged 22 to 72 years with known hypertension, dyslipidemia, or fasting hyperglycemia	160	Atkins (<20 g/day), Zone (macronutrient balance), Weight Watchers (calorie restriction), or Ornish (fat restriction) diet groups	12 months	No significant differences in BP found between the groups	Each popular diet modestly reduced body weight and several cardiac risk factors at 1 year.	Overall dietary adherence rates were low
Truby et al. [65]	2006	Randomized controlled trial	Adults in the United Kingdom	293	Dr Atkins’ new diet (<20 g/day), Slim-Fast plan, Weight Watchers pure points programme, and Rosemary Conley’s eat yourself slim diet and fitness plan	6 months	Low-CHO Diet (Atkins) resulted in SBP reduction at the end of the study of 7.20 mmHg and DBP mean reduction at the end of the study of 4.90. The reduction was not significantly different from the other diets	Regression analysis showed that total weight loss over time had the greatest influence on SBP and DBP	Compliance with each diet varied greatly
Gardner et al. [66]	2009	Randomized trial	Overweight premenopausal women	311	Atkins (<20 g/day) vs. Zone vs. LEARN vs. Ornish diet	20 months	The decrease in mean BP levels was largest in the Atkins group at all time points (at 20 months SBP mean difference −7.60 mmHg; DBP mean difference −4.40 mmHg)	More favourable control of other CV risk factor for low-CHO diet	Lack of valid assessment of individual compliance
Yancy et al. [67]	2010	Randomized controlled trial	Overweight or obese outpatients from the Department of Veterans Affairs primary care clinics	160	Low-CHO diet (< 20 g/day) vs. low-fat diet plus orlistat	48 weeks	Low-CHO group had a significantly larger reduction in SBP −5.9 mm Hg (95% CI, −8.8 to −3.1) and DBP −4.5 mm Hg (95% CI, −6.6 to −2.5) than low-fat + orlistat group (*p* < 0.001)	The 2 interventions resulted in similar weight loss	Reported not relevant
Lim et al. [68]	2010	Randomized controlled trial	Pts of 47 ± 10 years, BMI 32 ± 6 kg/m^2^ and one additional cardiovascular risk factor	113	Very low carbohydrate (10% of carbs) vs. very low fat (VLF) vs. high unsaturated fat (HUF) vs. control group (no intervention)	24 months	Decreases in body weight and DBP in the diet groups (−2.9 ± 5.2 mmHg) were significantly different vs. the control group (+0.8 ± 5.0) (*p* < 0.05)	Equivalent control of the cardiovascular risk factors with the three diets tested	Level of adherence reported as “modest”
Foster et al. [69]	2010	Randomized controlled trial	Pts of 45.5 ± 9.7 years and mean BMI of 36.1 ± 3.5 kg/m^2^	307	Low-CHO diet (20 g/day) vs. low-fat diet	24 months	SBP mean difference at the end of the study: −2.68 mmHg; DBP mean difference at the end of the study: −3.19 mmHg. No differences between groups at any time		32% of dropout in the low-fat group vs. 42% in the low-CHO
Liu et al. [70]	2013	Randomized controlled trial	Chinese women (30–65 years with a BMI = 24 kg/m^2^)	50	Low-CHO non-energy-restricted diet (20 g/day) vs. low-CHO (20 g/day) energy restricted	12 weeks	SBP mean difference at the end of the study −4.60 mmHg; DBP mean difference at the end of the study −2.70 mmHg; no difference between groups		Adherence reported 96%
Bazzano et al. [71]	2014	Randomized controlled trial	Men and women aged 22 to 75 years with a BMI of 30 to 45 kg/m^2^ without CV disease or diabetes	148	Low-CHO (<40 g/day) vs. low-fat diet (<30% fat; <7% saturated fat)	12 months	SBP mean difference at the end of the study −3.60 mmHg; DBP mean difference not reported. No significant difference between groups	Low-CHO diet was more effective for weight loss and cardiovascular risk factor reduction	Level of adherence reported as “high”
Cicero et al. [72]	2015	Multi-Center, Cross-Sectional Observational study	Outpatients aged 30–69 years with a BMI of 27–37 kg/m^2^ and abdominal circumference ≥ 98 cm for men and ≥ 87 cm for women	377	Very-low carbohydrate ketogenic diet (carbs 2–6 g/die), low fat, proteins intake 1.2/1.5 g/kg of ideal body weight	12 months	SBP (−10.5 ± 6.4 mmHg, *p* < 0.001) and DBP (−2.2 ± 3.1 mmHg, *p* < 0.001) changed from baseline to three months but no further changes were detected until the end of the study	The max reduction of body weight and CV risk factor levels was found after 12 weeks and only maintained after	66 patients dropped-out for different reasons, but none of which was a clinical matter

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
