# Peer review of "Ketogenic Diet, Physical Activity, and Hypertension—A Narrative Review"

_nutrients, 2021, doi:10.3390/nu13082567_

Round 1
Reviewer 1 Report
The introduction does not flow very well. The concepts are inaccurate and mixed: low carbohydrates and ketogenic diets are treated equally, thought these have very different metabolic effects. Short term vs long term, benefits vs risks, animal vs human studies. All these are intermingled in a way that it is hard to navigate, and it is not clear to the reader where the authors are heading. It does not set up the stage for the main analysis they want to report: effect of KD diets on exercise and BP.
Important comment: LCD are not ketogenic diets. In order to maintain ketosis, carbohydrates have to be very low, around 20 g per day; some might be able to tolerate a little more, but never above 50 g per day. You can’t talk about the benefits or risks of ketosis on a low carb diet, only on a ketogenic diet where you know the patient (or participant) is producing ketones.
Line 60- unclear what they mean by LCD have non-secondary effects (?) on CVD.
The Carbohydrate-Insulin Model described in paragraph 75-80 is not proven. A recent commentary by Speakman and Hall was published recently (Science 372 (6542), 577-578) where they discuss the complexity of insulin action.
Paragraph starting on line 81- not accurate explanation. I believe they meant the following: Both low carb and low fat (and thus high carb) diets when they are calorie-deficient, will mobilize fat stores and to some degree protein stores (including muscle tissue). Low carb diet will additionally deplete glycogen stores. Though they don’t mention anything about lipolysis, which is an important source of energy during caloric deficit (on any diet).
Line 103- statement that KD decrease visceral fat- this is not proven. The referenced paper was a study that was not only a KD but it was a very low calorie KD. Very low calorie diets do reduce visceral fat stores.
Line 109- Neither the South Beach diet nor the Atkins diets are ketogenic diets.
Line 110- Keto flu is mostly due to dehydration, mineral imbalance (more sodium is excreted during the KD diet) and it usually happens during the first week or two as the person transitions from using glucose to using ketones as a source of fuel.
Line 124- vegetables are not a good source of protein. More importantly to their point, would be to talk about plant vs animal sources of fat: fatty meats, cheese, cream vs olive oil, avocados, nuts.
Line 131- mentions that body weight might be increased by long term KD diet (from reference-12 weeks is long term in mice), but line 102: “Many are the long-term effects of the metabolic changes driven by KD. Among them, decrease in weight…”
Ketogenic diets and exercise section
Main idea of this section: KD diets might benefit endurance training as they would provide better access to fat stores for fuel. Resistance training, might benefit from higher carbohydrate diets as these rely on anaerobic production of energy.
The problem with this section is that the evidence from the studies does not support their point, especially the first one (KD diets and endurance training) as they themselves point out.
Figure makes sense, but this is not a new idea or concept.
Ketogenic diets and BP
This section, like the introduction, does not flow well and the message is not clear. The authors want to discuss effect of KD on BP (per the header and the title on the manuscript), but it is mixed with results on lipids, endothelial function, myocardial health, and other CVD risk factors.
They make a good case for all the limitation to the available data, and they conclude that the relationship between KD and BP is unclear.
Paragraph line 229- BP is one of the possible components of metabolic syndrome. The authors agree that the DASH diet was proven to be beneficial in managing BP, then they go on to say: “dietary regimens restricting carbohydrates seems to be more effective in controlling all MS features”. The DASH diet is not a diet restricted in carbohydrates.
Diets in table 1- maybe only 4 were ketogenic, the rest were low to moderate carbohydrate. Most of these studies would not be informative to answer the question posted.
Line 280- study referenced to show amelioration of carotid artery distensibility, was done in children 6-14 years old, and it would be important to see if this applies to adults who have stiffer arteries. Even in this study, there seemed to be only 4 out of the 13 studied children who improved in distensibility from 12 to 24 months.
Line 289- As mentioned above, I do not know of the evidence for a KD reducing visceral fat stores.
Paragraph line 291- the discussion of the effect of ketones on the myocardium is interesting and relevant to CVD (not necessarily BP). Line 303 mentions the short-chained fatty acids produced by the microbes by fermentation and this is also true and important, but the dots are not well connected. The microbes mostly ferment carbohydrates (as the authors themselves report), so the fact that the product of fermentation might benefit CVD, is in opposition to a ketogenic diet, which would not provide the required source for the microbes to produce these.
Conclusion:
Nice discussion of the limitation of the low carb diet studies and of the problems with diet adherence to a KD.
But basically the discussion says, correctly, that there are many confounders and limitations in the current literature to ascertain the effects, especially long term, of a ketogenic diet on exercise and blood pressure.
Some of the writing in this section is awkward and hard to follow. For example long sentence lines 336-341.
I did not understand what the authors meant on point 2, line 359. It seems to me to negate the point made in #1 right before that.
Line 362 claims that KD promote increase in lean mass. That is not substantiated. Maybe increase in percent lean mass as fat mass is reduced?
Author Response
REVIEWER #1
We would like to thank you for the detailed review of our manuscript. We greatly appreciate the effort you made concerning your critique for the review of our study. We have accepted all your suggestions and revised the article according to them.
The introduction does not flow very well. The concepts are inaccurate and mixed: low carbohydrates and ketogenic diets are treated equally, thought these have very different metabolic effects. Short term vs long term, benefits vs risks, animal vs human studies. All these are intermingled in a way that it is hard to navigate, and it is not clear to the reader where the authors are heading. It does not set up the stage for the main analysis they want to report: effect of KD diets on exercise and BP.
We would like to thank the reviewer for the very appropriate comment: the introduction has been completely rewritten according to the suggestions provided
comment: LCD are not ketogenic diets. In order to maintain ketosis, carbohydrates have to be very low, around 20 g per day; some might be able to tolerate a little more, but never above 50 g per day. You can’t talk about the benefits or risks of ketosis on a low carb diet, only on a ketogenic diet where you know the patient (or participant) is producing ketones.
Thank you for the valuable comment. Taking into account what we said in our introduction, ie, that there is no clear definition of "ketogenic diet", and that the suggested intake of carbohydrates in the various studies is variable, with definitions ranging from Low Carb Diet to Very Low Carb Diet, etc., we re-evaluated all the studies analyzed according to the daily intake of carbohydrates provided, and discussed mainly the studies in which this was an intake < 50 g per day of carbohydrates or less. Table 1 was modified accordingly.
Occasionally we have examined studies in which less ketogenic diets have been tested, but we have stressed any differences with the "orthodox" ketogenic diet to avoid misunderstandings.
Line 60- unclear what they mean by LCD have non-secondary effects (?) on CVD.
The sentence is not clear, it has been modified in the text by specifying that we refer to effects in some cases unfavorable
The Carbohydrate-Insulin Model described in paragraph 75-80 is not proven. A recent commentary by Speakman and Hall was published recently (Science 372 (6542), 577-578) where they discuss the complexity of insulin action.
Thanks for the suggestion. Given the unclear cause-and-effect relationship linking high-carbohydrate diet, hyperinsulinism, and obesity, as appropriately reported by the reviewer, this mechanism is no longer described in the revised version.
Paragraph starting on line 81- not accurate explanation. I believe they meant the following: Both low carb and low fat (and thus high carb) diets when they are calorie-deficient, will mobilize fat stores and to some degree protein stores (including muscle tissue). Low carb diet will additionally deplete glycogen stores. Though they don’t mention anything about lipolysis, which is an important source of energy during caloric deficit (on any diet).
Thank you for the suggestion; this paragraph has been entirely rewritten by clarifying the central role of storage lipid utilization
Line 103- statement that KD decrease visceral fat- this is not proven. The referenced paper was a study that was not only a KD but it was a very low calorie KD. Very low calorie diets do reduce visceral fat stores.
The paper by Valenzano, A.; et al. (Effects of Very Low Calorie Ketogenic Diet on the Orexinergic System, Visceral Adipose Tissue, and ROS Production. Antioxidants (Basel) 2019, 8, 643. doi: 10.3390/antiox8120643), tested a very low carbohydrate diet, in which patients were provided with <50 g of carbohydrate per day. The authors themselves, in the text state that " A very low carbohydrate diet is generally considered an equivalent of the ketogenic diet” confirming that to date there is much imprecision about the terms used. Nevertheless of these, the paragraph has been rewritten. We hope to have better defined the point in light of the available evidence. Regarding the finding of visceral fat reduction induced by KDs, actually some authors report a reduction in visceral fat mediated through the KDs as reported in the 2019 systematic review and consensus statement from the Italian Society of Endocrinology (SIE) ref #85 in our paper. Nevertheless, this uncertain finding is no longer hemphasized in the revised version.
Line 109- Neither the South Beach diet nor the Atkins diets are ketogenic diets.
Given the need to better define the examined scenario and avoid misunderstandings, the two diets are no longer mentioned in the text in the revised version.
Line 110- Keto flu is mostly due to dehydration, mineral imbalance (more sodium is excreted during the KD diet) and it usually happens during the first week or two as the person transitions from using glucose to using ketones as a source of fuel.
Thank you for the point, the paragraph has been completely rewritten in the revised version describing more properly what keto flu is and what are the main hypothesized causes for its occurrence
Line 124- vegetables are not a good source of protein. More importantly to their point, would be to talk about plant vs animal sources of fat: fatty meats, cheese, cream vs olive oil, avocados, nuts.
Very good point. We have made this concept more clear and precise in the revised version
Line 131- mentions that body weight might be increased by long term KD diet (from reference-12 weeks is long term in mice), but line 102: “Many are the long-term effects of the metabolic changes driven by KD. Among them, decrease in weight…”
Thank you for the suggestion. We have deleted this sentence. In the revised version we decided not to mention papers developed on animal models
Ketogenic diets and exercise section
Main idea of this section: KD diets might benefit endurance training as they would provide better access to fat stores for fuel. Resistance training, might benefit from higher carbohydrate diets as these rely on anaerobic production of energy.
The problem with this section is that the evidence from the studies does not support their point, especially the first one (KD diets and endurance training) as they themselves point out.
We would like to thank the reviewer for the comment: another very good point. The issue is now clearly discussed, addressed, and the discrepancy between the premises and the available data is stated.
Figure makes sense, but this is not a new idea or concept.
Thank you for the suggestion. The whole figure concept has been redesigned
Ketogenic diets and BP
This section, like the introduction, does not flow well and the message is not clear. The authors want to discuss effect of KD on BP (per the header and the title on the manuscript), but it is mixed with results on lipids, endothelial function, myocardial health, and other CVD risk factors.
They make a good case for all the limitation to the available data, and they conclude that the relationship between KD and BP is unclear.
Thanks for the suggestion. The paragraph has been rearranged to make the concept more adherent to the aim of the review, focusing more on the effects of KDs on blood pressure values, in comparison with other diets that are not (or less) ketogenic, and limiting the collateral analyses related to the effects on other elements affecting the cardiovascular risk. In the revised version is now reported also that the improvement in blood pressure values obtained through the diet is largely mediated by the reduction in body weight, fat mass and by the indirect effects resulting from the improved control of cardiovascular risk factors so make sense to briefly discuss the effects of KDs on other cardiovascular risk factors.
Paragraph line 229- BP is one of the possible components of metabolic syndrome. The authors agree that the DASH diet was proven to be beneficial in managing BP, then they go on to say: “dietary regimens restricting carbohydrates seems to be more effective in controlling all MS features”. The DASH diet is not a diet restricted in carbohydrates.
Thanks for the comment. In the revised version, we clarified that the sole intent of mentioning the DASH diet was to point out that a specific dietary regimen (not necessarily low-carb or ketogenic) can favorably impact blood pressure values
Diets in table 1- maybe only 4 were ketogenic, the rest were low to moderate carbohydrate. Most of these studies would not be informative to answer the question posted.
Table 1 was modified according to the reviewer’ suggestion. We re-evaluated all the studies analyzed according to the daily intake of carbohydrates provided by the various diets, and included only the studies in which the intake of carbohydrates was < 50 g per day or less.
Line 280- study referenced to show amelioration of carotid artery distensibility, was done in children 6-14 years old, and it would be important to see if this applies to adults who have stiffer arteries. Even in this study, there seemed to be only 4 out of the 13 studied children who improved in distensibility from 12 to 24 months.
We would like to thank the reviewer for the indication; given the limited usefulness of the study with reference to the objective of the review we decided not to cite it in the revised version.
Line 289- As mentioned above, I do not know of the evidence for a KD reducing visceral fat stores.
As we have replied before, some authors report a reduction in visceral fat mediated through the KDs (see ref #85 in our paper). Nevertheless, this uncertain finding is no longer hemphasized in the revised version.
Paragraph line 291- the discussion of the effect of ketones on the myocardium is interesting and relevant to CVD (not necessarily BP). Line 303 mentions the short-chained fatty acids produced by the microbes by fermentation and this is also true and important, but the dots are not well connected. The microbes mostly ferment carbohydrates (as the authors themselves report), so the fact that the product of fermentation might benefit CVD, is in opposition to a ketogenic diet, which would not provide the required source for the microbes to produce these.
We agree with the reviewer and the paragraph has been shortened in the revised version. However, we thought it was important to emphasize, although not directly related to hypertension, how recent therapeutic perspectives see ketones at the forefront of metabolic support to the failing heart.
Conclusion:
Nice discussion of the limitation of the low carb diet studies and of the problems with diet adherence to a KD.
But basically the discussion says, correctly, that there are many confounders and limitations in the current literature to ascertain the effects, especially long term, of a ketogenic diet on exercise and blood pressure.
Some of the writing in this section is awkward and hard to follow. For example long sentence lines 336-341.
Thanks for the suggestions. The conclusions have been lightened and made more readable
I did not understand what the authors meant on point 2, line 359. It seems to me to negate the point made in #1 right before that.
Actually points 1 and 2 are not in conflict with each other. The concept we were attempting to express is the following: the KDs reduce blood pressure values but do not guarantee a greater effect on the control of tension values than non-ketogenic diets
Line 362 claims that KD promote increase in lean mass. That is not substantiated. Maybe increase in percent lean mass as fat mass is reduced?
KDs do not increase lean mass. This statement has been corrected in the revised version.
We hope that we have successfully changed our manuscript according to your suggestions and that we have provided all the necessary explanations. We also hope that the manuscript now fulfills your criteria, and the Journal criteria for publication.
Reviewer 2 Report
The topic of the review is very timely and relevant. However, the authors included in their review both ketogenic and low-carb diets with up to more than 35% of carbohydrates. This is not appropriate as the latter do not induce the production of ketones and have different metabolic, vascular, and exercise-related effects compared to a ketogenic diet. As a result, the conclusions of the review are of little utility and potentially misleading. The authors should consider excluding the studies that tested low carb non-ketogenic diets. In addition, figure 1 is of very poor quality from both a conceptual and design point of view. The authors should consider using clearer graphics and descriptions of the underlying biochemical pathways.
Author Response
REVIEWER #2:
We would like to thank you for your expert review of our manuscript. Thank you very much for your positive opinion regarding our manuscript. We put a lot effort in this study and we appreciate your opinion very much.
The topic of the review is very timely and relevant. However, the authors included in their review both ketogenic and low-carb diets with up to more than 35% of carbohydrates. This is not appropriate as the latter do not induce the production of ketones and have different metabolic, vascular, and exercise-related effects compared to a ketogenic diet. As a result, the conclusions of the review are of little utility and potentially misleading. The authors should consider excluding the studies that tested low carb non-ketogenic diets. In addition, figure 1 is of very poor quality from both a conceptual and design point of view. The authors should consider using clearer graphics and descriptions of the underlying biochemical pathways.
Thank you for the valuable comment. Taking into account what we said in our introduction, ie, that there is no clear definition of "ketogenic diet", and that the suggested intake of carbohydrates in the various studies is variable, with definitions ranging from Low Carb Diet to Very Low Carb Diet, etc., we re-evaluated all the studies analyzed according to the daily intake of carbohydrates provided, and discussed mainly the studies in which this was an intake < 50 g per day of carbohydrates or less. Table 1 was modified accordingly.
The whole figure 1 concept has been redesigned and improved both in graphics and in the text message.
We hope that we have successfully changed our manuscript according to your suggestions and that we have provided all the necessary explanations. We also hope that the manuscript now fulfills your criteria, and the Journal criteria for publication.
Round 2
Reviewer 1 Report
Abstract:
Lines 11-12: a healthy lifestyle is no different than one to improve CVD factors. The way it is written, it makes it seem as if these are different ways to lower CVD risk.
Line 17: no strong evidence for a ketogenic diet improving insulin resistance
Introduction:
Line 49: change word “determine ketosis” to “increase production of ketones”
Line 58: change “but not from red blood cell” to “but not for red blood cells”
Line 76: you can’t say a maximum of 20g and then follow up with “a range of 20-50 g”
Line 94: why are you mentioning here the risk of low calorie diets as a potential risk of CVD?
Line 108: a main source of energy is from dietary fat, not just stored fat. Also, not a clear explanation of how ketone bodies are produced.
Paragraph around lines 141: it is not the plant vs animal proteins that affect LDL levels, but rather plant vs animal fats.
Ketogenic diet and Physical Activity:
Line 226: why do you say that the metabolic adaptation seen in the athletes can be extended to less trained subjects and to diets different than those tested?
The current manuscript is more precise and evidenced based.
Author Response
REVIEWER #1
We would like to thank you for this second review of our manuscript and for your comments to further improve the quality of our work. We greatly appreciate the effort you made concerning your critique for the review of our study. We have accepted all your suggestions and then revised the article according to them.
Abstract:
Lines 11-12: a healthy lifestyle is no different than one to improve CVD factors. The way it is written, it makes it seem as if these are different ways to lower CVD risk.
Line 17: no strong evidence for a ketogenic diet improving insulin resistance
Thanks for the comments. We have better clarified in the abstract the role of a healthy lifestyle in reducing cardiovascular risk and the role of the ketogenic diet in glucose metabolism.
Introduction:
Line 49: change word “determine ketosis” to “increase production of ketones”
Thank you. We have edited the sentence according to your suggestion.
Line 58: change “but not from red blood cell” to “but not for red blood cells”
Thank you. We have edited the sentence according to your suggestion.
Line 76: you can’t say a maximum of 20g and then follow up with “a range of 20-50 g”
Thank you for your comment. We have made it clearer in the revised version that “we can talk of KD for a diet which provides a < 50 g per day of carbohydrates (equivalent to a percentage < 10%) although very often only a daily intake < 20 g is allowed”
Line 94: why are you mentioning here the risk of low calorie diets as a potential risk of CVD?
Thank you for your comment. The idea we were trying to express is the need to study aimed to more in-depth assess the short-term and long-term effects exerted by LCDs and KDs in order to determine whether this diets may be safely implemented in patients at high CV risk. The structure of the sentence has been changed and hopefully is now clearer.
Line 108: a main source of energy is from dietary fat, not just stored fat. Also, not a clear explanation of how ketone bodies are produced.
Thanks for the comment. In the revised version it is explained that storage fat is used as soon as the availability from dietary incoming fat is spent. Biochemistry of ketone bodies is also improved.
Paragraph around lines 141: it is not the plant vs animal proteins that affect LDL levels, but rather plant vs animal fats.
Thanks for the comment. In the revised version we have made clearer that the LDL level resulting from the ketogenic diet depends on the type of lipids consumed and not on the proteins, but the study by Fung et al that we are citing here shows that the origin of the proteins consumed (vegetable or animal) is also an item to consider because it affects for example cardiovascular mortality.
Ketogenic diet and Physical Activity:
Line 226: why do you say that the metabolic adaptation seen in the athletes can be extended to less trained subjects and to diets different than those tested?
Indeed, ours is a speculation based on plausibility. We added in the revised version the statement "it is likely that they can be extended to less trained subjects and to even more ketogenic diets, although suitable experimental confirmations are needed to substantiate the magnitude of the metabolic adaptation in other groups of subjects”.
The current manuscript is more precise and evidenced based.
We hope that we have successfully changed our manuscript according to your suggestions and that we have provided all the necessary explanations. We also hope that the manuscript now fulfills your criteria, and the Journal criteria for publication.
Reviewer 2 Report
Thank you for making these changes, which have highly improved the manuscript quality.
Author Response
REVIEWER #2:
Thank you for making these changes, which have highly improved the manuscript quality.
We would like to thank you again for your expert review of our manuscript and the quality of your suggestions, we are glad that we have provided all the necessary explanations and that the manuscript now fulfills your criteria for publication.